# Biological Application of the Allopathic Characteristics of the Genus *Maclura*: A Review

**DOI:** 10.3390/plants12193480

**Published:** 2023-10-05

**Authors:** Juan Carlos Sainz-Hernández, Edgar Omar Rueda-Puente, Yaeel Isbeth Cornejo-Ramírez, Ariadna Thalía Bernal-Mercado, Héctor Abelardo González-Ocampo, Betzabe Ebenhezer López-Corona

**Affiliations:** 1Centro Interdisciplinario de Investigación para el Desarrollo Integral Regional, Unidad Sinaloa, Instituto Politécnico Nacional, Sinaloa 81049, Mexico; jsainz@ipn.mx; 2Departamento de Agricultura y Ganadería, Universidad de Sonora, Blvd. Luis Encinas y Rosales, Sinaloa 83323, Mexico; edgar.rueda@unison.mx; 3Departamento de Investigación y Posgrado en Alimentos, Universidad de Sonora, Blvd. Luis Encinas y Rosales, Sinaloa 83000, Mexico; yaeel.cornejo@unison.mx (Y.I.C.-R.); thalia.bernal@unison.mx (A.T.B.-M.)

**Keywords:** antioxidant, antimicrobial, flavonoids, phenolic compounds, morin

## Abstract

*Maclura* is a plant genus little known and used, species of which have been mainly used in the recovery of soils, for medicinal purposes such as dental infection treatments, and as wood for making furniture and construction. The overexploitation of this genus has placed certain species in endangered extinction status in some countries, such as Brazil. In addition, the scarce research and information limit the development, cultivation, and management of its species regarding their biochemical composition, which includes bioactive compounds such as the phenolic and flavonoid compounds found in some species such as *M. pomifera*, *M. cochinchinensis*, and *M. tinctoria*. The plants’ antioxidant, antimicrobial, anticancer, anti-inflammatory, and antiproliferative activities have been attributed to these compounds. Other biochemical components such as ashes, insoluble lignin, holocellulose, and the high content of lipids and carbohydrates have been identified to be used to produce biofuels, with characteristics very similar to fuels derived from petroleum. This review aims to analyze the current knowledge on the plant genus *Maclura*, exploring its biochemical compounds and potential applications, including as a food additive, antioxidant supplement, in agriculture, for therapeutic purposes, aquaculture, and the cosmetic and industrial sector.

## 1. Introduction

*Maclura* spp. is a plant genus that is taxonomically classified (Table 1) within the *Moraceae* family that is widely distributed throughout the world; it is found on practically all continents, except Antarctica, it includes trees and shrubs and some species of lianas, it is made up of 63 genera and more than 1500 species in the world, and it is found mainly in tropical regions.

The genus *Maclura* was named after the renowned American geologist William Maclure and consists of 40 species with a tropical distribution from North America to South America [1,2] and in many Asian countries, such as China, Japan, Korea, Taiwan, India and Thailand [3]. It is the most widely distributed genus in the *Moraceae* family and is the only one with spines [4]. 

The *Maclura* spp. presents distichus, unequal leaves, with male flowers in spikes and female flowers in heads [4,5]; its seeds are oval-elongated with a size between 8 and 12 mm, its fruits (syncarps) are similar to those of other *Moraceae*, such as *Morus* and *Broussonetia*, being usually the size of a grapefruit with a fleshy consistency that begins to appear between 12 and 15 years of age. Some are edible and they have a high amount of pulp (*M. tricuspidata*, *M. africana* and *M. cochinchinensis*), and their reproduction and dispersal since prehistoric times has been through the digestive tract of the fauna that consumed them, mainly extinct species such as the mammoth and the giant sloth.

The *Maclura* tree requires, for its development, a well-drained soil rich in organic matter, which can range from deep soils to those close to estuaries. The seeds must be handled almost immediately after extracting the fruit, washing and separating the pulp and then drying it for 72 h to sow immediately or refrigerate. A viability period for the seed of only three months has been reported, so it should be handled as soon as possible. Before planting, the area that must be shaded or covered must be selected to avoid the transfer of sunlight. It is important to disinfect beforehand against damping-off, and soak the seed for a period of 12 h. Finally, sand-seed must be mixed and distributed homogeneously in the selected area. Watering must be carried out lightly daily. Since the plant does not tolerate water flooding, waterlogging should be avoided and some type of drainage available. It is recommended to eliminate the lower branches of the tree to provide greater support and shape. Lastly, some rooting promoters such as acid indole butyric acid (IBA) and naphthalene acetic acid (NAA) have been used to improve plant growth and development [6].

The *Maclura* spp. has been widely used in traditional medicine due to its biochemical characteristics (Figure 1) [3,7,8,9]. The species of this genus are a rich source of bioactive compounds, mainly flavonoids as the principal constituents [1,10,11,12,13,14]. For example, *Maclura pomífera*, one of the most studied species, contains isoflavones in the fruits, root, bark, and heartwood of the plant [13,14,15,16,17]. The substantial presence of phenolic compounds within the bark and the occurrence of flavonoids and dibenzophenones within the leaves confer upon it a spectrum of beneficial attributes encompassing anti-inflammatory, antimicrobial, antioxidant, anticancer, and antiproliferative activities [1,13,18,19,20,21,22,23,24].

Furthermore, the species belonging to the *Maclura* genus have demonstrated pharmacological potential in addressing dental infections [25] and combating conditions such as gout, hyperuricemia, and inflammation [11]. Traditionally, the leaves of this plant are used as bandages after tooth extractions to reduce pain and swelling [8]. The extracts of *M. aurantiaca* show antiprotozoal capacity, demonstrating high inhibitory efficiency against leishmaniasis in in vitro studies [26]. In addition, lipid extracts of *M. pomifera* have been shown to regulate the prostatic index, improving biochemical markers and prostate function in rats with sulpiride-induced prostatic hyperplasia [27].

On the other hand, *Maclura* species constitute a rich source of pigments that hold promising potential for utilization as dyes in the food and textile industries. The wood and leaves of the *Maclura* spp. contain a high content of pigments such as morin, maclurina, and morina that can be isolated from their bark and used for dyeing wool, textiles, silk, and leather [3,28,29,30]. Furthermore, the extract of *Maclura* species can be used for obtaining biomaterials combined with other polymers. For example, a phenolic extract of *M. tinctoria* heartwood was added to form structured aerogels based on nanofribilated cellulose to produce a biomaterial for medical applications [19].

In addition to the biological importance of *Maclura*, the plant is viewed as a viable alternative to promote reforestation, since it is very long-lived and has attributes that allow it to be used in processes of ecological restoration and soil recovery, as well as in association with of other agricultural crops. Currently, climate change on the planet forces us to seek reforestation alternatives to mitigate the increase in temperature and reduce greenhouse gas emissions, for which reason species such as *Maclura* represent a very valuable ecological alternative.

*Maclura* plants can develop under a wide variety of environmental conditions; this allows the distribution of various species such as *M. tinctoria*, *M. pomífera*, *M. cochinchinensis*, and *M. tricuspidata*, which are among the few species of the genus which will be studied and described in this review.

## 2. *Maclura tinctoria*

*Maclura tinctoria* (L) D. Don ex Steud is also known as *Morus tinctoria* L., *Morus alba*, *Chlorophora*, *Xanthoxylum*, and *Chlorophora tinctoria*. It develops mainly in humid soils and along slopes in Mexico, and it is commonly called “moral”, “nail moral” “dinde”, or “moral stick”. It is a leafy, thorny, milky, medium-sized tree growing from 10 to 30 m tall with a trunk 50 to 100 cm wide, a density of 0.83 g/cm^3^, and highly resistant to attack by organisms [31,32]. The phenology of *M. tinctoria* begins with its emergence from 10 to 20 days after the seeds are sown. Although the germination percentage is relatively low, at approximately 30%, it is important to note that the seeds do not demonstrate any dormancy [33]. Its flowering occurs from September–October, exhibiting its ripe fruit from December–January [31]. This fruit is usually juicy, sweet, and has a good flavor, is rich in seeds [2,34], is generally consumed in countries like Brazil (raw, in juice, or desserts) and its dispersal is carried out mainly by bats and local aerial fauna.

The scarce research and information on *M. tinctoria* limit species development, cultivation, and management. Although extensive literature exists on species such as *M. pomífera* and *M. cochinchinensis*, the study on *M. tinctoria* is minimal. The overexploitation of the species in the last 30 years has caused it to be considered endangered in Brazil [33]. In addition, the plant has a low germination rate in its seeds, which has not been thoroughly studied, and it does not allow the reproduction of new phenotypes of the species. Some investigations have shown that the application of growth promoters and some phytohormones, such as ANA (naphthylacetic acid) and AIB (indolbutyric acid), in *M. tinctoria* favor the germination process and significantly increase root development, as well as the number of shoots, and improve plant survival [35]. Possibly, the use of compounds with biostimulant activity, including chitosan and some microorganisms (*Azotobacter*, *Azospirillum*, *Pseudomonas*, *Acetobacter*, *Burkholderia*, and *Bacillus*), may favor the development of *M. tinctoria* by increasing the availability of nutrients and stimulating the synthesis of phytohormones, improving rooting and foliage production, as has been observed in other plant species like corn [36], tobacco [37] rice [38], orchid [39], and *Salicornia bigelovii* [40].

*Maclura tinctoria* contains a high amount of phenolic compounds (Figure 2), including epicatechin (65.80 mg/L), catechin (47.05 mg/L), gallic acid (40.80 mg/L), quercetin (25.22 mg/L), syringaldehyde (1.71 mg/L), ferulic acid (0.42 mg/L), p-hydroxybenzoic acid (0.81 mg/L), and syringic acid (0.10 mg/L) as the major compounds [1,19,20,41,42,43,44]. It has a basic density and weighted density of 0.54 g cm^−^³ and 0.55 g cm^−^³, respectively, and contains 1.21% of ashes and a relatively low content for insoluble lignin (17% to 19%), 71.02% holocellulose, and 12.62% total extracts. Based on the chemical content, the species is unsuitable for producing cellulosic pulp but can be used for other purposes [1,45].

In the fruit of *M. tinctoria*, we identified, via HPLC analysis, thirty-three phenolic compounds, of which nine are phenolic acids (198.05 mg GAE g^−1^), nine are flavonols (78.57 mg EQ g^−1^) (two methylquercetin-3-glucosides, two kaemferol-3-O-glucosides, and four quercetin-3-O-glucosides), four are dihydro-flavonoids, four are flavones (one apigenin glycoside, two luteolin ide, and one chrysoeriol glycoside), and seven are unidentified compounds [46].

Its bark contains the highest amount of polyphenols compared to other plant parts. The leaves also present many polyphenols, notably a morin flavonoid [18]. Morin is a widely studied yellowish pigment [47,48,49,50] which belongs to the class of flavonols and has also been detected in *M. cochinchinensis* [3,29,30], in the shell of the almond (*Prunus dulcis*), in the leaves, bark, fruits, and other vegetative parts of the *Chlorophora tinctoria*, Indian guava leaves (*Psidium guajava*), and in onion, apple, and coffee [50]. Morin (Figure 3) has antioxidant properties [50] that have been determined in *Maclura* spp. The wood and bark of *M. tinctoria* show a high content of phenolic compounds and high antioxidant activity [1], corroborating the idea that antioxidant activity is proportional to the content of phenolic compounds [51,52,53,54].

The study of *M. tinctoria* is very recent and scarce, mainly identifying its phenolic compounds, proanthocyanidin, its antioxidant activity, and its antimicrobial activity from extracts obtained from the bark and wood (25 μg/mL) [1,13,46]. *M. tinctoria* has antimicrobial activity at minimum inhibitory concentrations (MIC) in a wide range of bacteria, including *Streptococcus sanguinis*, *S. mutans*, *S. mitis*, *Prevotella nigrescens*, *Actinomyces naeslundii*, *Porphyromonas gingivalis*, *S. mutans*, *Porphyromonas gingivalis*, *Chromobacterium violaceum*, *S. oralis*, *Enterococcus faecalis*, *Staphylococcus aureus*, *Escherichia coli*, and *Aeromonas hydrophila* in fish cultures [1,19,20,25].

Secondary metabolites of a phenolic nature are responsible for the biological activity in plants [55,56]; specifically, in *M. tinctoria*, a high content of phenolic compounds has been demonstrated, especially tannins (43.8 mg·L^−1^) and flavonoids (108.74 mg·L^−1^) [19], which have even been used in other areas such as aquaculture, showing a positive effect on the growth of juvenile shrimp (*Lithopenaeus vannamei*) grown in the laboratory and improving the response of immune system markers to pathogens in the postlarval stage. This suggests the potential of the plant to formulate diets of nutritional quality and in the reduction of losses due to diseases in aquaculture [46].

Other studies [44] show the potential of the species *M. tinctoria* and *M. pomífera* on the HIV virus, where several prenylated xanthones were isolated and characterized, including macluraxanthona B, macluraxanthona C, and dihydrocudraflavona B, which showed potential results for the anti-HIV test, pointing to the plant as a viable candidate in the treatment and prevention of diseases such as HIV-AIDS.

## 3. *Maclura pomífera* (Raf.) Schneid

Another of the best-known species of this genus is *M. pomífera* (Raf.) Schneid, also known as Osage orange. It is found mainly in areas with large plains [57], generally in the United States and some parts of Canada [58]. It has a short growth period in which it grows from 9 to 12 m in height. It is usually long-lived, living up to 150 years or more [58,59]. The fruit of *M. pomífera* is yellow to greenish in color, circular in shape, with a diameter that measures 10 to 15 cm, similar to a grapefruit with a thick and heavy texture [60]. It is filled with fiber and sticky latex and its evolution occurred as a consequence of the extinction of the main dispersing species such as mammoths, which made it an anachronistic species adapted to the available dispersal. The seeds of its fruit represent 11% of its weight and are composed mainly of protein (33.9%) and fat (32%), followed by carbohydrates (20.8%), ash (6.7%), and water (5.9%) [61].

It has recently been the subject of research since it has been found to contain a high content of lipids and carbohydrates that can be used to produce biofuels, even with characteristics very similar to petroleum-derived fuels [58,62,63]. Additionally, the fruit, bark, leaves, root, and seed have been reported to have a high content of oils, sugars, and compounds such as isoflavones, xanthones, triterpenes, and stilbenes, with isoflavones being the most representative [15].

These compounds have antioxidant, antimicrobial, anti-inflammatory, and antitumor biological activity [4,24]. In this way, its fruit has traditionally been used as an insect repellent, as a source of pigment, and even for medicinal purposes for toothache and cancer treatments [4,63]. Despite this, human intake is not recommended since toxic elements such as lead, arsenic, and mercury have been found in all fruit regions in values above the permissible limits established for food [59].

The extraction process of the active compounds from the *M. pomífera* plant has a direct effect on the quality and quantity of these, in addition to the fact that it is a difficult process to carry out due to the adhesive nature of the *Maclura* compounds, especially in the condensation stage, where the extracts adhere to the solvent to be evaporated, which causes high losses and low extraction yields. One of the methods proposed to solve this problem is high hydrostatic pressure, a non-thermal process that can be effectively used to extract active compounds of various polarities [60]. This technique could represent an effective alternative to reduce the losses of bioactive compounds of interest in the species *M. pomífera*, improving their yields and quality; however, further studies are required.

The fruit of *M. pomífera* has shown an effect on human cancer cells (kidney, lung, prostate, breast, melanoma, and colon) as an inhibitor of histone deacetylase (HDAC) via the prenylated flavonoid pomiferin, showing antiproliferative activity in the six cell lines evaluated [23]. In this sense, the compound pomiferin (contained in the fruit of *M. pomífera*) has been shown in tests with cancer cells to behave as an inhibitor of cancer stem cells from a human glioma [22], showing a reduction in the expression of genes associated with stamina (the ability of the cell to reproduce repeatedly and form stem cells). Also, *M. pomífera* has been helpful as a marker in the diagnosis of cancer since it allows the distinguishing of patients with prostate cancer from those patients who present benign prostatic diseases and normal subjects, this being through the high affinity of the sera of patients with prostate cancer towards the *M. pomífera* lectin [21].

*Maclura pomífera* has also been used in nanotechnology through the formulation of silver nanoparticles, in which *Maclura* was used as a reducing agent and stabilizer, obtaining well-structured, uniform, and spherical nanoparticles. These nanoparticles also showed antimicrobial activity against Gram-negative bacteria, which suggests a bioactive potential in medicine for the treatment of diseases associated with this type of microorganisms [64]. Likewise, the secondary effect on the total weight of testicles of male Sprague Dawley rats treated with *M. pomífera* for 5 days has been reported, finding an increase in weight but also a decrease in sperm mortality. This suggests a limited benefit in the reproductive parameters of the rats studied; however, further studies are required using longer periods, with more days of treatment and higher concentrations of *Maclura* [65].

The growth and survival of plant species in forests and tropical areas, such as *Maclura*, is a current problem that is caused by the effect of droughts, stress conditions, and environmental changes. By the end of the 21st century, the outlook is expected to worsen due to the increase in temperature due to climate change, for which the selection of plants tolerant to drought and severe stress conditions will be necessary. The species of *Maclura pomífera* is a species of tree that could be an alternative to this problem when cultivated to promote reforestation, thanks to the fact that it is a plant that is tolerant of drought and the stress that it causes [57].

## 4. *Maclura cochinchinensis* (Lour.) Corner

*Maclura Cochinchinensis* is a woody climbing plant with thorns 0.5 to 2.5 cm long, with elliptical-shaped leaves that do not exceed 8 cm long. Its fruits are round, yellow-orange in color, and its heartwood has generally been used mainly as a dye and medicine in the Asian regions where it is native. It is found naturally distributed mainly in countries such as Asia, India, Japan, China, Indochina, Malaysia, and Thailand [3]. It is commonly called rooster thorn or “Kae Lae”, and within its chemical composition, the presence of compounds such as morin, resveratrol (Figure 4), and hydroxyresveratrol has been evidenced [11,66]. In addition, it has been reported to have a variety of biological activities, such as antioxidant and anti-inflammatory [11,67], antibacterial against strains *of Staphylococcus aureus*, *S. epidermidis*, *Bacillus subtilis* [3,10], *Enterococcus faecalis*, *Micrococcus luteus*, *S. pyogenes*, *Shigella flexneri*, and *Salmonella Typhimurium*, fungicidal against *Candida albicans* [10], and antiviral against Herpes simplex virus [68].

In Eastern countries, its traditional use has focused on treating medical skin disorders, rheumatism, and hepatitis [3,68,69]. Thai culture has, for many years, included the use of *M. cochinchinensis* as a medicine against skin diseases, diarrhea, chronic fever, and lymph node abnormalities. In the cochinchinensis species, the presence of morin has been determined as one of the most abundant compounds, so its presence in *Maclura* suggests its high potential as an antioxidant, anti-inflammatory, and anticancer source [70].

Previous works have demonstrated the anti-inflammatory and hyperuricemic capacity of the heartwood of the plant in vitro by evaluating the enzymatic kinetics and in vivo in hyperuricemic mice induced by potassium oxonate (PO), pointing to *M. cochinchinensis* as a promising candidate in the natural treatment for inflammation and hyperuricemia that causes gout [11].

Another important component found in high amounts in *M. cochinchinensis* is resveratrol, a substance present in several plant species such as grapes, blueberries, peanuts, and blackberries. Its presence is mainly associated with the plant’s response to various stress conditions [71]; resveratrol is a natural polyphenol of great importance in health due to its biological properties such as antioxidant, anticancer, antiaging, antiplatelet aggregation, anti-inflammatory, antiallergic, antiobesity, antidiabetes, cardioprotectivity, and neuroprotectivity [72,73]

Studies such as that of Lakornwong et al. [74] have identified the cytotoxic potential of *M. cochinchinensis*, managing to isolate three furanoxanthones, macochinxanthones, and sixteen xanthones from its roots that exhibited potent cytotoxicity against four cancer cell lines (KB, HelaS3, A549, and HepG2). In this sense, it has been identified that the prenyl group located in position C-6 of prenylisoflavones 1–4 and 6–7 improved the cytotoxicity of the isoflavone against the KB and HepG2 lines [75]. In this way, species of the genus *Maclura* have been used for cancer treatment due to their in vitro inhibitory effect on highly metastatic B16F10 melanoma cells [76]. The results demonstrated that the active extracts of n-hexane and chloroform, as well as macluraxanthone and geronthoxanthone-I isolated from *M. amboinensis*, exhibited potent antiproliferative effects, possibly through the induction of apoptosis of highly metastatic B16F10 melanoma cells. It was noted that *Maclura* may represent a new chemopreventive and/or chemotherapeutic agent in cancer prevention and/or chemotherapy.

For their part, Vongsak et al. [67] used different solvents such as ethanol and methanol and techniques such as ultrasound to improve the extraction process and quality of bioactive compounds in *M. cochinchinensis*, where ethanol and methanol extracts 80% presented better antioxidant activity under the DPPH and ABTS technique; with respect to ultrasound, this technique improved the antioxidant and antityrosinase capacity, for which it was concluded that the extracts of *M. cochinchinensis* obtained via these methods can be potentially used for the manufacture of cosmetic products and as a marker for quality control.

*Maclura cochinchinensis* can also be used in the textile sector: Griyanitasari et al. [77] evaluated three plant extracts, obtained from water, ethanol, and water-ethanol, on tilapia skin as a tanning agent as an alternative for the elimination of chrome (the most widely used agent in the leather industry), finding that the *Maclura* species immediately produced a unique tan color to the fish; the aqueous extract was the one that presented the best results in the variables analyzed such as tensile strength and strength. This represents an ecological alternative for the substitution of chemical agents such as aluminum and formaldehyde that cause severe damage to the environment.

## 5. *Maclura tricuspidata*

*Maclura tricuspidata Carrière* (traditionally called *Cudrania tricuspidata*) is a medium-sized plant that can grow up to 7 m high, with thorns 0.5 to 2 cm long; its bark is brown in color, and in winter they are stained with a reddish in color; the leaf of this species is oval from 3 to 6 cm long; it has its flowering period in May–June and its fruit in June–July. It is a species that has been used for years in the traditional medicine of Korea and other East Asian countries such as China and Japan, where its bark and root have been used in the treatment of gastrointestinal tumors and for gonorrhea, rheumatism, jaundice, scabies, bruising, and dysmenorrhea [78].

It is cultivated mainly from the seed; however, the growth period can last up to ten years, so it is necessary to sow the seed immediately after extracting it from the fruit. An alternative used for the species is the micropropagation technique via cuttings, a very effective method for reproducing difficult-to-cultivate plant species with low germination percentage and slow growth, which allows the plant to be obtained in large quantities and in a shorter period of time, in addition to the use of growth-promoting products such as indole acid butyric acid (IBA), indole acetic acid (IAA), or some phytohormones that greatly improve the growth variables and physiology of the plant.

The fruit of *M. tricuspidate* is edible; its particular red color makes it attractive for consumption. Its flavor is usually slightly insipid, but once it reaches maturity, it resembles a watermelon flavor, contains a high amount of sugar, and has about six small seeds. It has been reported to have antioxidant and anti-inflammatory properties thanks to its high content of xanthones, flavonoids, and benzenoids [79].

Choi et al., in 2019 [80], evaluated, in vitro, the effect of *M. tricuspidata* on the enzymes pancreatic lipase, α-amylase, β-glucosidase, phosphodiesterase IV, alkaline phosphatase, and citrate synthase, all related to obesity. The assays evidenced the inhibitory capacity of these phenolic compounds towards these enzymes, suggesting that the compounds detected from the *M. tricuspidata* fruit extract may regulate carbohydrate/lipid/energy metabolism by inhibiting obesity-related enzymes. Likewise, different bioactive compounds from the fruit of *M. tricuspidata* have been identified via spectroscopic and chemical methods, showing neuroprotective effects against cell death induced by 6-hydroxydopamine (6-OHDA) in SH-SY5Y cells of human neuroblastoma [81]. These investigations indicate the potential of the fruit of *Maclura* spp. as an important source of bioactive molecules; however, it is necessary to carry out more studies on these products’ chemical composition, mechanism of action, and biological activity.

Park et al. [82], determined the antimetastatic properties of *M. tricuspidata* root, stem, leaf, and fruit extracts with high phenol and flavonoid content and antioxidant capacity in hepatocellular carcinoma cells, showing that these extracts can be used in potential treatment for hepatocellular carcinoma with the possibility of improving therapeutic efficiency. Regarding the antimicrobial activity of *M. tricuspidata*, its effect on *Streptococcus iniae* has been studied [83], showing high inhibition and evidencing the role of prenylation of isoflavones from fruit and leaf extracts on its antibacterial activity.

*Maclura tricuspidata* has also been studied for its effect in the treatment of diseases such as Alzheimer’s and Parkinson’s, with findings indicating that the plant decreases the damage caused by oxidative stress in cells and increases the expression of genes that encode the antioxidant enzymes superoxide dismutase (SOD) and catalase (CAT), producing a protective and beneficial effect against neurodegeneration [84].

For years, the species has been used in Asian regions due to the biological characteristics of its extracts; however, it has been mentioned that the method of obtaining it is of great importance for the quantity and quality of the bioactive compounds. Some techniques, such as liquid chromatography high-efficiency HPLC (High-Performance Liquid Chromatography) and reverse-phase high-performance liquid chromatography (RP-HPLC), have been used for their identification and quantification, indicating the presence of components of great biological interest found in greater quantity in the premature stages of the fruit, concluding that, at this stage, the fruit of *M. tricuspidata* is ideal to be considered as a raw material in the production of functional foods [12].

On the other hand, it has been reported that the essential oil obtained from *M. tricuspidata* contains abundant functional biological properties. In the study carried out by Yong et al. [85], the chemical composition and antioxidant capacity of the oil obtained from fully ripe fruits was determined with the steam distillation technique, identifying 76 compounds such as fatty acids and their esters, p-cresol (393.5 μg/100 g dw^3^), δ-cadinene (147.17 μg/100 g dw^3^), β-caryophyllene (145.7 μg/100 g dw^3^), n-nonanal (140.3 μg/100 g dw^3^), teaspiran A (121.3 μg/100 g dw^3^), and teaspiran B (99.67 μg/100 g dw^3^), among others, and a high antioxidant activity in the fraction glycosidically linked; this suggests the potential of the species in the cosmetic, pharmaceutical, and edible oils industries; however, more studies are required.

## 6. Conclusions

*Maclura* spp. is a genus of little-used plants, the limited research and information on the genus has limited its development, cultivation, and management. The use of species of the *Maclura* genus promises a potential source of compounds with biological activity of great interest for nutraceuticals, health, and industry. The presence of phytochemical components such as phenolic compounds and flavonoids has indicated high antioxidant, antimicrobial, antiproliferative, anti-inflammatory, antiviral, and anticancer activity. Compounds such as morin and resveratrol have been identified as one of the main components of the genus, demonstrating its high biological capacity and making it an ideal candidate for the treatment of neurodegenerative diseases such as Parkinson’s and Alzheimer’s. Likewise, and thanks to its biochemical characteristics, it can potentially be used as a food additive, an antioxidant supplement, for the development of nutraceutical products, for therapeutic and agricultural purposes, and even in the aquaculture and cosmetics sectors. *Maclura* is a species considered in danger of extinction in South American countries such as Brazil due to the overuse of its wood. Studies such as this one promotes the ecological recovery of the species and point out the importance of its components beyond timber and construction purposes and show the presence of a very important amount of compounds such as phenols and flavonoids that must be addressed in future research as part of the search to improve extraction techniques and for the effective isolation of these in greater quantity and quality.

## Figures and Tables

**Figure 1 plants-12-03480-f001:**
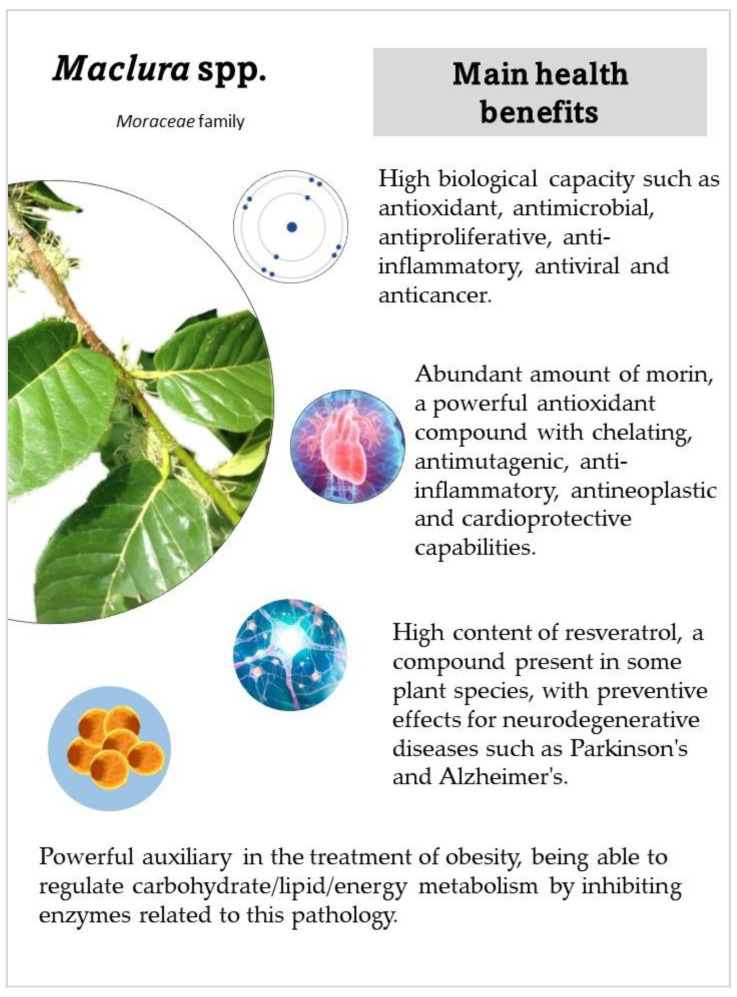
Main health benefits of *Maclura* spp. Own elaboration.

**Figure 2 plants-12-03480-f002:**
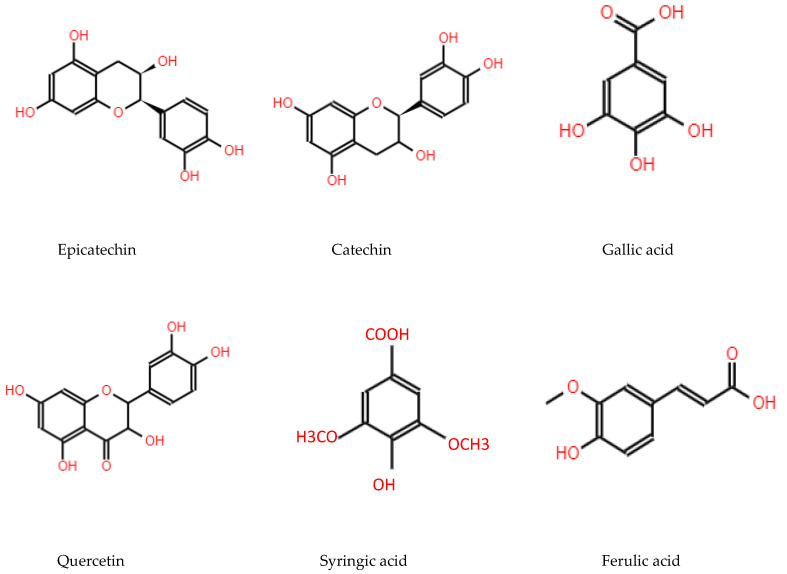
Chemical structure of the main phenolic compounds in *M. tinctoria*, own elaboration, using the JSME Molecular Editor program.

**Figure 3 plants-12-03480-f003:**
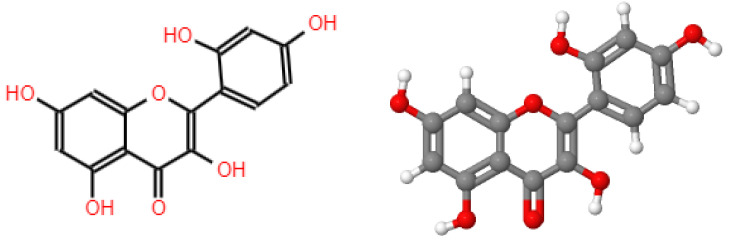
Chemical and 3D structure of the morin, own elaboration, using the JSME Molecular Editor program.

**Figure 4 plants-12-03480-f004:**
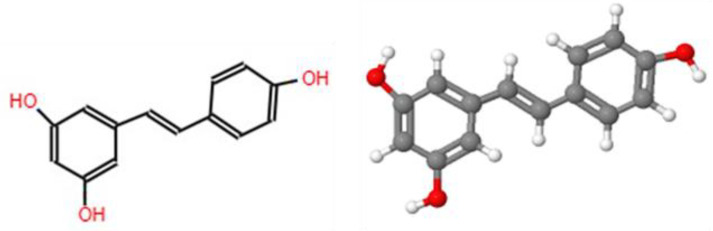
Chemical and 3D structure of resveratrol, own elaboration, using the JSME Molecular Editor program.

**Table 1 plants-12-03480-t001:** Taxonomic classification of *Maclura* spp.

Kingdom	*Plantae*
*Phylum*	*Magnoliophyta*
Class	*Magnoliopsida*
Order	*Urticales*
Family	*Moraceae*
Genus	*Maclura*

## Data Availability

Not applicable.

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
