# Peer review of "Biological Application of the Allopathic Characteristics of the Genus Maclura: A Review"

_plants, 2023, doi:10.3390/plants12193480_

Round 1

Reviewer 1 Report

Dear authors, your article is very interesting, clear and new, congrats!

I only recommend a few things, like:

1)         Mention the ideal conditions of growth

2)         Line 46: “cochinchinensis) and They have a high amount of pulp,” rearrange this sentence.

3)         Lines 85 and 112: “Maclura” at the beginning of sentences -> revise all the article

4)         Insert the chemical structures of the predominant compounds

5)         Lines 140 to 143 and 231 to 234: bacteria names in italic

6)         Line 146 and 153 and others (please revise the all article): “M. tinctoria” and “M. pomífera“ in italic

7)         This issue can be further explored, I recommend the following:

a.                     You present the plants and insert a new subsection within each sp. to stand out their health benefits, in vivo and in vitro

8)         Insert an image about their health-benefits

9)         Values about the most promising phytochemicals and micro and macronutrients found

10)       A deepen discussion is also recommended.

Minor editing of English language required

Author Response

We have carefully read the comments and suggestions from the section editor’s and from the reviewers. We really want to express our deep thankfulness for the reviewers and for you.

REVIEWER 1

  1. Mention the ideal conditions of growth

Response: An extensive paragraph was written on the ideal conditions and management of the cultivation of the Maclura tree: The Maclura tree requires, for its development, a well-drained soil rich in organic matter, which can range from deep soils to those close to estuaries. The seeds must be handled almost immediately after extracting the fruit, washing and separating the pulp and then drying it for 72 hours to sow immediately or refrigerate, a viability period for the seeds has been reported of only three months, so it should be handled as soon as possible. Before planting, the area that must be shaded or covered must be selected to avoid the transfer of sunlight. It is important to disinfect beforehand against damping-off, and soak the seed for a period of 12 hours. Finally, sand-seed must be mixed and distributed homogeneously in the selected area. Watering must be carried out lightly daily. Since the plant does not tolerate water flooding, waterlogging should be avoided and some type of drainage available. It is recommended to eliminate the lower branches of the tree to provide greater support and shape. Lastly, some rooting promoters such as acid Indole butyric acid (IBA) and naphthalene acetic acid (NAA) have been used to improve plant growth and development.

  1. Line 46: “cochinchinensis) and They have a high amount of pulp,” rearrange this sentence.

Response: The wording of the text requested by the reviewer was corrected: The Maclura spp. presents distichus, unequal leaves, with male flowers in spikes and female flowers in heads [4, 5], its seeds are oval-elongated with a size between 8 and 12 mm, its fruits (syncarps) are similar to those of other Moraceae, such as Morus and Broussonetia, it is usually the size of a grapefruit with a fleshy consistency that begins to appear between 12 and 15 years of age, some are edible and they have ahigh amount of pulp (M. tricuspidata, M. Africana and M. cochinchinensis) and They have a high amount of pulp, and their reproduction and dispersal since prehistoric times has been through the digestive tract of the fauna that consumed them, mainly extinct species such as the mammoth and the giant sloth.

  1. Lines 85 and 112: “Maclura” at the beginning of sentences -> review all the article

Response: Maclura abbreviation at the beginning of several paragraphs was corrected, writing the entire word.

  1. Lines 140 to 143 and 231 to 234: bacteria names in italic

Response: Several figures were made presenting the main bioactive compounds of the genus Maclura.

  1. Lines 140 to 143 and 231 to 234: bacteria names in italic

Response: The italic format was established for the scientific names of species and bacteria throughout the manuscript.

  1. Line 146 and 153 and others (please review the all article): “M. tinctoria” and “M. pomifera“ in italic

Response: The italic format was established for the scientific names of species and bacteria throughout the manuscript.

  1. This issue can be further explored, I recommend the following: to. You present the plants and insert a new subsection within each sp. to stand out their health benefits, in vivo and in vitro

Response: In the current original writing, several existing in vitro and in vivo studies for the genus Maclura are described, however, the information on this genus is very limited, therefore, having a section for each species on these two options is very complicated. However, the original writing does include in written form the available in vitro and in vivo scientific information.

  1. Insert an image about their health-benefits

Response: Various figures were included on the main bioactive compounds present in the Maclura genus.

  1. Values about the most promising phytochemicals and micro and macronutrients found

Response: The values of the main phytochemicals described throughout the manuscript were included.

  1. A deep discussion is also recommended.

Response: The discussion was extended into different paragraphs of the manuscript, which have been adapted according to the reviewer's kind comments.

Reviewer 2 Report

The authors did very well work reviewing on Biological application of the allopathic characteristics of the genus Maclura: a review.

Please add in the manuscript the separate medicinal values of the Maclura genus. 

Write a conclusion of the future potential of these plants for useful society and human purposes, focused solely on phytochemical ingredients, and also will included this being significant for the medical worth of these plants.  

No

Author Response

September 20, 2023. Sonora, Mexico

Dr. José Antonio Morales-González

jmorales101@yahoo.com.mx

Editor-in-Chief

Dra. Nancy Vargas Mendoza

nvargasmendoza@gmail.com

ccp. Dr. Steven Tian steven.tian@mdpi.com

According the Manuscript, entitled:

Biological application of the allopathic characteristics of the genus Maclura: a review which we submitted to Plants, has been changed considering the comments of the reviewer(s).

We have carefully read the comments and suggestions from the section editor’s and from the reviewers. We really want to express our deep thankfulness for the reviewers and for you.

We truly are indebted to all of you for valuable help to improve the paper.

We are sending to you two attachments:

1.- ARTICLE FINAL VERSION (Manuscript Maclura plants new version)

2.- Document in word (plants-2623374 modifications made)  –answer to all observations INDICATING IN different colour THE CHANGES INTO THE DOCUMENTO.

3.- In this document, we are answering all observation. Please continue below to see the tables of

   responses to editor, reviewer 1 and reviewer 2

The authors hope this satisfy the standard to be accepted by the editorial advisory board.

Thank you for your consideration and looking forward to hearing from your soon again, I remain.

Sincerely yours,

AUTHOR

REVIEWER 2

No.

Reviewer Comment

Change made

1

Please add in the manuscript the separate medicinal values of the Maclura genus.

Values were established for the main bioactive components described in the article.

2

Write a conclusion of the future potential of these plants for useful society and human purposes, focused solely on phytochemical ingredients, and will also include this being significant for the medical value of these plants.

The conclusion of the article was modified according to the reviewer's recommendations, including future research studies, the importance of this plant in health, the environment and humanity and highlighting the phytochemical importance of the genus.

Maclura spp. it is a genus of very little used plant, its limited research and information has limited its development, cultivation and management. The use of species of the Maclura genus promises a potential source of compounds with biological activity of great interest for nutraceuticals, health and industry. The presence of phytochemical components such as phenolic compounds and flavonoids have shown high antioxidant, antimicrobial, antiproliferative, anti-inflammatory, antiviral and anticancer activity. Compounds such as morin and resveratrol have been identified as one of the main com-ponents of the genus, demonstrating its high biological capacity and making it an ideal candidate for the treatment of neurodegenerative diseases such as Parkinson's and Alzheimer's. Likewise, and thanks to its biochemical characteristics, it can potentially be used as a food additive, antioxidant supplement, for the development of nutraceutical products, for therapeutic and agricultural purposes, and even in the aquaculture and cosmetic sec- tor. Maclura is a species considered in danger of extinction in South American countries such as Brazil due to the overuse of its wood. Studies such as this one promotes the eco-logical recovery of the species and point out the importance of its components beyond timber and construction purposes and show the presence of a very important amount of compounds such as phenols and flavonoids that must be addressed in future research in the search to improve extraction techniques and for the effective isolation of these in greater quantity and quality.

Round 2

Reviewer 1 Report

Dear Authors,

Please, insert the name of the program in which you made the chemical structures and include images regarding the health benefits of Maclura and their spp. 

Author Response

Dear Dr Reviewer 1,
We attach a new version considering the latest observations considered by you:
1.- the name of the program in which the chemical was performed.
2.- structures and includes images about the health benefits of Maclura and its spp.

Apologies are offered for the incidents previously caused. respectful greetings
